# Gastrointestinal Stromal Tumors: Our Ten-Year Experience of a Single-Center Tertiary Hospital

**DOI:** 10.3390/jpm13081254

**Published:** 2023-08-13

**Authors:** Georgios Tzikos, Alexandra-Eleftheria Menni, Despoina Krokou, Angeliki Vouchara, Soultana Doutsini, Eleni Karlafti, Anestis Karakatsanis, Aristeidis Ioannidis, Stavros Panidis, Theodosios Papavramidis, Antonios Michalopoulos, Daniel Paramythiotis

**Affiliations:** 11st Propedeutic Department of Surgery, AHEPA University Hospital, Aristotle University of Thessaloniki, 54636 Thessaloniki, Greece; alexandra.menni@gmail.com (A.-E.M.); despik@windowslive.com (D.K.); aggvou9@gmail.com (A.V.); taniadoutsini@gmail.com (S.D.); ankarakatsanis@gmail.com (A.K.); ariioann@yahoo.gr (A.I.); st.panidis@gmail.com (S.P.); papavramidis@hotmail.com (T.P.); amichal@auth.gr (A.M.); danosprx1@hotmail.com (D.P.); 2Emergency Department, AHEPA University General Hospital of Thessaloniki, 54636 Thessaloniki, Greece; linakarlafti@hotmail.com

**Keywords:** gastrointestinal stromal tumors, GIST, case series, review, Greece

## Abstract

Background: Gastrointestinal stromal tumors (GISTs) are the most frequent mesenchymal neoplasms of the gastrointestinal tract. They have variable clinical presentation, prognosis, and molecular characteristics. Here, we present the results of our retrospective study including patients operated on for GIST during the last decade. Methods: All the patients who underwent GIST resection during the decade 2008–2018 were included in the study. The diagnosis was based on the pathology report. All the data were collected and analyzed statistically using the Statistical Package for Social Science v25.0. Finally, after having applied the proper search terms, a comprehensive review of articles published in the Medline database was held. Results: Thirty-two patients (sixteen women) were included in the study with a mean age of 69.6 years old (SD = 13.9). Twenty-one patients had a GIST in the stomach, eight in the small intestine, and three had an extra GIST. Of the 29 patients contacted, 21 were alive with a mean survival time of 74.3 months (SD = 49.6 months, min: 3.0 months, max: 161.0 months), whereas eight patients passed away. Finally, 13 patients were treated with tyrosine kinase inhibitors (TKIs) of whom only one died, while 9 patients passed away from those treated with surgery alone (*p* = 0.031). Conclusions: Our results were in concordance with the existing data in the literature. GISTs require patient-based therapeutical management depending on the histology of the tumors. Gastric tumors present a better prognosis than those localized in the intestine, while the use of TKIs has led to an improvement in patient survival rate.

## 1. Introduction

Gastrointestinal stromal tumors (GISTs) are the most frequent mesenchymal neoplasms of the gastrointestinal tract and, in general, they are highly resistant to conventional chemotherapy and radiotherapy. Most but not all GISTs are characterized by gain-of-function KIT (75–80%) or PDGFRa (5–10%) gene mutations. GISTs without mutations in KIT or PDGFRa, also called wild-type GISTs, have different clinical and molecular characteristics and are usually associated with syndromes such as Carney’s Triad, Neurofibromatosis type I, and Carney–Stratakis syndrome [1]. GISTs are typically present in adults over 40 years (median age 55–60 years) and only exceptionally in children. Most GISTs occur in young adults, and in children, most are wild-type. They can present anywhere in the gastrointestinal (GI) tract from the lower esophagus to the anus and in an ectopic location outside the GI tract, which are also known as extra-GISTs (E-GISTs). Histologically, GISTs vary from cellular spindle cell tumors to epithelioid and pleomorphic ones, the morphology differs somewhat by site. By definition, GISTs are KIT(CD117)-positive. Positivity for nestin (90–100%) and CD34 (70%) are also characteristic but less specific features. Other markers such as the cell cycle marker ki-67 or PCNA are used for the prognosis of the patients, the increased expression of whom has been associated with a less favorable prognosis.

Predictive factors for malignancy are a mitotic rate over 5 per 50 HPF or a size over 5 cm [2,3,4]. These tumors are more common among males (rate ration (RR) = 1.35) and in black populations than in white populations (RR = 2.07). Their incidence increases with age and its peak occurs during the eighth decade of life [5]. In the Greek population, there are limited data regarding the prevalence of GISTs which is estimated at approximately 140 new cases per year, with a mean age of occurrence during the sixth decade of life and a small male predominance (male/female = 1.4/1) [6]. The aim of this study is to present the experience of a single-center tertiary hospital in dealing with GISTs during the last decade and a review of the current literature on this field.

## 2. Materials and Methods

We have retrospectively searched all the records according to the pathology reports of our patients who have undergone surgery in our institution during the decade 2008–2018. We decided to examine this period because we aimed at a follow-up of at least 3 years based on the fact that in patients with response to the preoperative use of Imatinib, the recommendation is to continue the therapy with tyrosine kinase inhibitors (TKIs) for 3 years after the surgical resection [7]. All the cases with the diagnosis of “Gastrointestinal Stromal Tumor” based on histopathological and immunohistochemical analysis were included in the study. Then, every eligible patient’s pathology report was obtained, and the documented data, the demographics, and the clinical features including details about adjuvant therapy with TKIs were collected, analyzed, and recorded as well. Moreover, we contacted the patients or their relatives on the telephone in order to address any issue, including survival, during the postoperative period from the day of surgery until the time this study was initiated. The type of surgical procedure was determined by the location, the size, and the expansion of the tumor, and it was carried out after consensus by the surgical team.

Moreover, we conducted a comprehensive search for articles published in the Medline database. Only articles written in English were included. The search was performed using the following Mesh terms: “Gastrointestinal Stromal Tumor” or “Gastrointestinal Stromal Neoplasm”. Moreover, the titles and abstracts of all the studies identified were screened and assessed. The studies deemed suitable were thoroughly reviewed for eligibility according to their clinical relevance. An ultimate check of databases was carried out on 31 March 2022.

Regarding the statistical methods performed to analyze the collected data, the Shapiro–Wilk test was used to assess the normality of the data’s distribution, since the sample was <50 patients. The baseline characteristics were summarized using appropriate descriptive statistics. The results are presented as the mean ± standard deviation (SD), where normality was confirmed, and as the median and interquartile range (IQR) when the data were skewed. The Chi-square test was used to efficiently determine whether there was any association between categorical variables. The Kaplan–Meier method was used for the survival analysis. A *p*-value of <0.05 was considered statistically significant. Statistical analysis was performed with the Statistical Package for Social Science (SPSS), Inc. (v 25.0; Chicago, IL, USA).

## 3. Results

### 3.1. Patients

Thirty-two cases, identified with the diagnosis of GIST, were included in the study. Half of them were women, and the mean age was 69.6 years old (SD = 13.9 years) at the time of surgery. Twenty-one patients presented with GIST located in the stomach (62.5%) intra- or extraluminally (Figure 1 and Figure 2).

In eight patients (25.0%), the tumor was found in the small bowel, whereas one case (3.1%) was identified to have multiple GIST of the jejunum. Finally, in three cases (9.4%), an extra-GIST was reported (one in the left ischiorectal fossa, one in the omentum, and one in the pelvis).

### 3.2. Macroscopic, Histological, and Immunohistochemical Characteristics of the Tumor

The median tumor size was 4.75 cm (IQR = 5.25 cm, min: 0.5 cm, max: 15.3 cm). In three patients, microscopic tumor invasion of the resection margin was reported after the pathology analysis was performed. In addition, lymph nodes positive for metastasis were observed only in one case (3.13%).

Moreover, the mitotic rate (MR) varied among the cases. In 17 of them, the MR was less than or equal to 5/50 high-power fields (HPFs), whereas, in 10 reports, MR was more than 10/50 HPFs.

The expression of the CD117 antigen was detected in all the cases, while CD34 was reported in 28 patients. Positivity in other antigens such as S100 protein and smooth-muscle actin (SMA) varied. The detailed data are reported in Table 1.

### 3.3. Types of Procedure

Different types of procedures were held based on the location of the neoplasm. More precisely, for GISTs located in the stomach, local or partial gastrectomy was performed, while for those located in the small intestine, segmental resection was carried out. Furthermore, for E-GISTs, the tumor was excised en bloc. It should be mentioned that in one case, the en bloc resection of the stomach and a small part of the left lobe of the liver was performed (Figure 3), while only in another one did recurrence occur in the stomach after local resection, and the patient needed to undergo another partial gastrectomy for the resection of the recurrent tumor. Interestingly, in four cases, the procedure was held in the setting of urgent surgery, due to complications of the presenting GISTs (two cases were presented with ileus, one with upper GI hemorrhage, and one with small bowel rupture). One patient treated urgently due to hemorrhage was admitted to the ICU and remained there for 17 days. For the rest of the patients, no other major peri-procedural adverse events were documented, and all the other patients’ postoperative courses were uncomplicated.

### 3.4. Surveillance

Of the patients contacted, three did not answer our call. Moreover, 19 were found alive with a mean survival time of 74.3 months (SD = 48.8 months, min: 3.0 months, max: 161.0 months), whereas 10 patients passed away with a median survival time of 13.5 months (IQR = 49 months, min: 0.5 months, max: 84.0 months) (*p* = 0.19). Thirteen patients were treated with adjuvant therapy including TKIs, 12 with Imatinib, and 1 with Sunitinib, due to serious adverse events from the initially administered Imatinib. The response to the therapy was assessed with the help of repeated abdominal or pelvis CT scans every 6 months for the first 5 years. The dose for Imatinib was 400 mg daily for 3 years, and for Sunitinib 50 mg q.d., for 4 weeks; two cycles with a 2-week drug-free interval. Nine out of the thirteen patients completed the course of the therapy. The adverse events which were referred by the patient treated with Sunitinib were anemia and serious diarrhea when initially treated with Imatinib. For the rest of the patients who received Imatinib, only nonserious adverse events were recorded; two patients were referred with mild gastrointestinal disorders, and three other patients had anorexia and weakness. Moreover, from the patients who received TKIs, only one died, while from the patients treated with surgery alone, nine deaths were reported (*p* = 0.017). Of the patients who died, only in two cases did the relatives state that the cause was the recurrence of the GIST, while one passed away due to intracerebral hemorrhage and three more due to myocardial infarction. Moreover, GIST was located in the stomach in 8 out of 10 patients who died and in the small intestine in the other 2 patients. Finally, as it was expected, there was a significant difference (*p* = 0.021) in the survival between the patients treated with adjuvant therapy postoperatively and the patients treated with surgery alone (Figure 4).

## 4. Discussion

Historically, GISTs as a separate entity were firstly described by Mazur and Clark in 1983, who distinguished them from gastrointestinal smooth-muscle and nerve sheath tumors [8]. A few years later, in 1998, Kindblom et al. reported that the origin of these tumors arose from pluripotential mesenchymal stem cells which differentiate into the interstitial cell of Cajal [9]. The cells of Cajal are present in the myenteric plexus as well as between the smooth muscle fibers of the muscularis propria, functioning as pacemaker cells in intestinal peristalsis. The relationship of GISTs to Cajal intermediate cells was initially based on the fact that both tumor cells and Cajal cells expressed the transmembrane KIT receptor. However, the critical characteristic that distinguished GISTs from other stromal tumors was the discovery of mutations in the c-KIT proto-oncogene isolated in these neoplasms [10].

In the vast majority of patients, the initial clinical presentation of GISTs is nonspecific, such as the presence of vague abdominal discomfort or fullness, which can explain the often-noticed delay in diagnosis. Moreover, since GISTs arise from the submucosa, they may grow extraluminally without causing patients any symptoms, and as a result, GISTs could reach a very large size before causing more characteristic manifestations. In our case series, the greatest tumor was 15.3 cm in diameter, and it caused obvious abdominal distension (Figure 3), although the patient did not refer to any other symptom except this deformity. In addition, except in the four cases which were presented as emergencies due to complications caused by the tumor, in all the other cases, no specific symptoms were reported, and the tumor was discovered either accidentally during imaging studies and endoscopic or surgical procedures for irrelevant reasons or in the setting of general investigation for reported atypical symptoms. In most cases, the tumor’s presentation depends on its size, location, and growth pattern. It is reported that the median tumor size of GISTs that are detected incidentally is around 2.7 cm, whereas those presenting with symptoms have a diameter of 8.9 cm [11]. The mean diameter of GISTs in the literature ranges between 10 and 13 cm, and GISTs larger than 5 cm are more likely to cause symptoms [12]. In our case series, the median size of the GIST was 4.75 cm in diameter lower than the literature-reported diameter, while the smallest tumor was 0.5 cm. When patients experience symptoms, GISTs are presented mainly as GI bleeding. Bleeding may range from occult chronic bleeding (resulting in chronic anemia) to life-threatening melena or hematemesis. In our series, we reported only one case presenting with acute bleeding from the upper GI, in which an urgent gastrectomy was performed. GISTs may also present themselves as a palpable mass, as intestinal obstruction due to intraluminal growth or luminal compression, or as acute abdomen with pneumoperitoneum [13]. In our study, two cases underwent surgery due to obstruction, and in one more, the tumor was found after urgent laparotomy, due to intestinal perforation and peritonitis.

Concerning the prevalence and prognosis of the GISTs, data on worldwide frequency are limited, but in general, GISTs constitute 1–3% of all gastric malignancies. According to some population-based studies, the annual incidence of GISTs ranges from 6.5 to 14.5 cases per million [14,15]. Data from SEER (Surveillance, Epidemiology, and End Results Program), collected between 2001 and 2011, report that the 5-year overall survival (OS) for patients with GISTs is 77% for those with localized disease, 64% for regional-disease patients, and 41% for those with metastases, at the time of diagnosis. Furthermore, gastric GISTs have a relatively better prognosis than small bowel ones, even for those with a similar size or mitotic rate [16].

As mentioned above, the treatment of GISTs is largely related to the location of the tumor, its size, and mitotic activity [3]. This is because, based on the above characteristics, the tumors show variable biological behaviors and subsequent clinical prognoses. Regarding the clinical course and the possibility of recurrence, the larger the size of the tumor and the mitotic activity, the more frequent the occurrence of recurrence regardless of the anatomical location [17]. Table 2 lists the biological characteristics associated with the risk of aggressive behavior of GISTs, as Miettinen and Lasota reported in 2006 [18]. Patients with tumors characterized by large size and a mitotic index of >5 high-power field (HPF) appear to be at high risk of recurrence regardless of the tumor localization area [19].

On the other hand, tumors being smaller in size and having a lower mitotic index (<5 HPF) show better biological behavior, and they are less likely to recur. However, tumors that develop in the jejunum, ileum, duodenum, and rectum present a high risk of recurrence, regardless of having a small number of mitoses (<5 HPF). It should be mentioned that in the categorization of Miettinen and Lasota, the location of the tumor is taken under consideration for the risk assessment, and this is the main difference from the risk stratification proposed by Fletcher in 2002 [20]. In 2009, Gold et al. from the Memorial Sloan-Kettering Cancer Center created a nomogram that uses the size of the tumor, its mitotic rate, and its location to predict the 2-year and 5-year relapse-free survival after the resection of a localized stromal tumor [17]. The National Comprehensive Cancer Network (NCCN) criteria for the malignancy risk of primary GISTs have not been incorporated into the American Joint Committee on Cancer (AJCC) staging system yet. However, the NCCN staging criteria might be more effective in determining the individual risk for progressive disease after a complete R0 resection with a negative tumor margin [7].

Regarding imaging studies, a variety of techniques are used to detect the presence of a GIST. Plain abdominal radiography is nonspecific, and it is usually ordered in the emergency department for the diagnosis of a tumor complication, such as an obstruction or a perforation [21]. In double-contrast series (barium swallow or barium enema), GISTs appear as a filling defect that is demarcated and sharply elevated compared with the surrounding mucosa [22]. Ultrasonography is not an optimal choice for GIST imaging, while a computed tomography (CT) scan with contrast material is a necessary and very guiding imaging tool for GISTs diagnosis. It can also assist in the evaluation of possible metastatic disease and as a result contributes to tumor staging before treatment [23,24]. Furthermore, magnetic resonance imaging (MRI) is a helpful adjunct to CT, especially for large tumors with necrotic and hemorrhagic components. Particularly, GIST lesions have low intensity on T1 images and high intensity on T2 images, with mass enhancement after intravenous gadolinium administration. Finally, a PET scan (18 FDG PET) is recommended for the detection of metastatic lesions in patients with GISTs, but it is also used in advance to evaluate the tumor response to adjuvant therapy [25]. Usually, GISTs can be lost in the endoscopic control, due to their extraluminal or submucosal growth. It is worth noting, that for very small gastric GISTs < 2 cm, an endoscopic ultrasound-guided fine-needle aspiration biopsy (EUS-FNAB) is a diagnostic option [7].

In the majority, these tumors appear with high cellularity and spindle-shaped cells, even though these histological findings are not uniform. Ultrastructural parameters reveal a variety of both myoid and neural characteristics. Data investigated by immunochemical analysis and electron microscopy ended in the expression of various antigens, such as Nestin (90–100% positivity), CD34 (70% positivity; it is correlated with higher malignant behavior), CD44 (that usually demonstrates a better prognosis), Vimentin, Desmin, S100 protein, and other ones, most importantly, the expression of the CD117/c-kit protein. This antigen has a predominant role in the diagnosis of GISTs. In particular, CD117 is a growth factor receptor with tyrosine kinase activity and is a product of the proto-oncogene c-kit [26]. The most important factor is that this antigen facilitates the differential diagnosis of GISTs because it is expressed only in them and not in other smooth muscle or neural tumors [27,28].

The radical surgical excision of the tumor is the definitive treatment of choice for patients with localized or other potential resectable tumors. However, GISTs tend to grow extraluminally and generally tend not to infiltrate but to displace adjacent organs. Additionally, because in some cases, tumors are fragile if they rupture, there is a possibility of peritoneal dispersion [7]. These characteristics of GISTs, as well as their location, determine how the surgery is performed. Moreover, the laparoscopic resection of tumors is usually performed in tumors up to 5 cm in size [29,30,31]. Additionally, given the rarity of nodal metastases of the GISTs, lymphadenectomy is not usually mandatory, and the surgical therapy aims to obtain negative margins after the tumor resection [7,32].

Regarding medical treatment, it primarily consists of TKIs with mesylated Imatinib, the most commonly administered agent. Imatinib (2-phenylpyrimidine) is the main active ingredient in Gleevec (US) or Glivec (EU). This substance was developed as a drug in the late 1990s by Novartis Pharmaceuticals for the treatment of various types of chronic myeloid leukemia (CML). However, only in 2002, Imatinib received approval from the US Food and Drug Administration (FDA) for the treatment of advanced or metastatic GIST [33]. This accelerated approval for adjuvant uses in patients with resectable GIST, while in 2008, it was granted for accelerated approval as an adjuvant treatment following surgical removal [31]. Imatinib is a very potent inhibitor of tyrosine kinases such as ABL kinase, chimeric oncoprotein BCR-ABL of chronic myeloid leukemia, transmembrane KIT receptors, PDGFRα, and PDGFRb. The mechanism of this substance is essentially the inhibition of the binding of adenosine triphosphate (ATP) to the kinase residue. The results after Imatinib treatment in patients with GISTs are spectacular as disease control is observed in 70–85% of patients with advanced disease (median survival 36 months), the total response to treatment reaches 4–5%, while partial response with a reduction in tumor size occurs in 47–67% of patients. Moreover, in advanced tumors, Imatinib administration preoperatively facilitates surgical resection as it results in tumor shrinkage [7,34].

Before the approval of the tyrosine kinase inhibitor (TKI) Imatinib and other similar inhibitors as a treatment choice for the adjuvant therapy of the patients. The therapeutic strategy was based solely on total tumor resection. The period that we evaluated was between 2008 and 2018, immediately after the initial FDA approval of Imatinib, and, therefore, only 13 out of 32 patients received adjuvant therapy with TKIs. This relatively low number of patients who received Imatinib arose from the fact that TKIs were proposed for adjuvant therapy postoperatively after oncological consultation, which was based on the criteria of the study of Joensuu et al. [35]. However, as was expected, there was a significant difference (*p* = 0.017) regarding death occurring in favor of the patients having received TKIs. Today, in about 50% of patients, it is possible to be cured with surgery (with free microscopic resection limits, RO) without additional treatment [36]. However, in patients with metastatic disease, the additional use of adjuvant therapy with Imatinib or other inhibitors is necessary [37]. Stromal tumors are generally characterized by hepatic and peritoneal metastases, while lymph node metastases have only been reported in rare cases.

Over the years, new medical treatments have been added to the adjuvant therapy of GISTs. A major drawback that was discovered in recent years is the development of secondary mutations in patients resulting in tolerance to Imatinib. Patients develop tolerance to the drug during the first line of treatment. However, the problem of Imatinib tolerance is now bypassed by the use of other tyrosine kinase inhibitors such as sunitinib. Sunitinib was suggested by the FDA in 2006 in cases of progressive disease. In 2013, Regorafenib received FDA approval for locally advanced GISTs that had been considered unresectable [38]. Additionally, Avapritinib was approved for the PDGFRA mutations (including D842V mutation) in the treatment of patients with advanced GISTs [39]. In general, more and more drugs are being developed to replace Imatinib and sunitinib to cover the different resistance mechanisms that arise each time. Additionally, many side effects and adverse events have been reported during therapy with TKIs [40]. Although, in our sample, major events occurred only for one patient.

The strength of this study is that the data were collected from a tertiary hospital that covers an area with a population of about 2 million (20% of the nation’s population) and thus it adds significant information regarding the clinical entity of GISTs referring to the Greek population, which is quite small. However, the major limitations of this study are that it is a retrospective study and information bias is unavoidable. Therefore, the results should be evaluated carefully. Finally, the relatively small sample size was not suitable for further statistical analysis.

## 5. Conclusions

In this study, we described the experience of our tertiary surgical department on the diagnosis, evaluation, and treatment of GISTs. GISTs represent a diverse range of tumors which are mainly found along the GI tract but can rarely be found in an extra-gastrointestinal area. The majority of our cases were located in the stomach, and we dealt with three extra-GISTs as well. The prognosis of the disease is variable and depends on specific prognostic factors, the most significant of which are the size of the tumor, the number of mitoses, and the location of the neoplasm. Surgical therapy focusing on the complete resection of the tumor is the definite therapy, while medical therapy with TKIs as an adjuvant or even as a palliative choice of treatment ameliorates patient survival and quality of life.

## Figures and Tables

**Figure 1 jpm-13-01254-f001:**
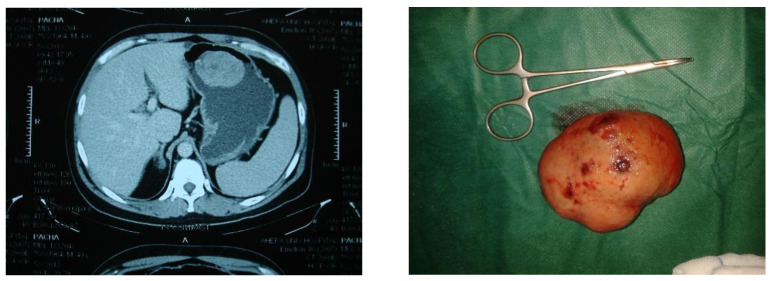
A case of an intraluminal GIST in the stomach. The tumor was excised after gastrotomy.

**Figure 2 jpm-13-01254-f002:**
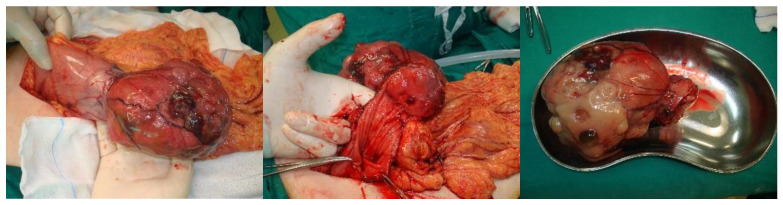
A case of an extraluminal GIST in the stomach. The tumor was excised via segmental gastrectomy.

**Figure 3 jpm-13-01254-f003:**
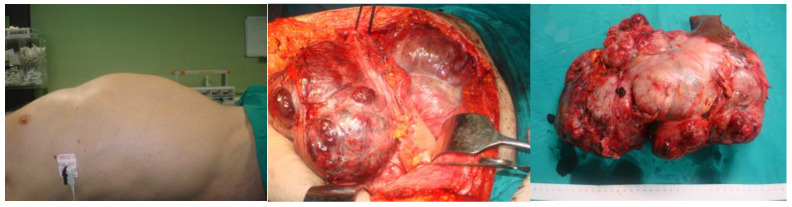
A case of a giant GIST in the stomach. Partial gastrectomy and en bloc resection of a small part of the left lobe of the liver were performed.

**Figure 4 jpm-13-01254-f004:**
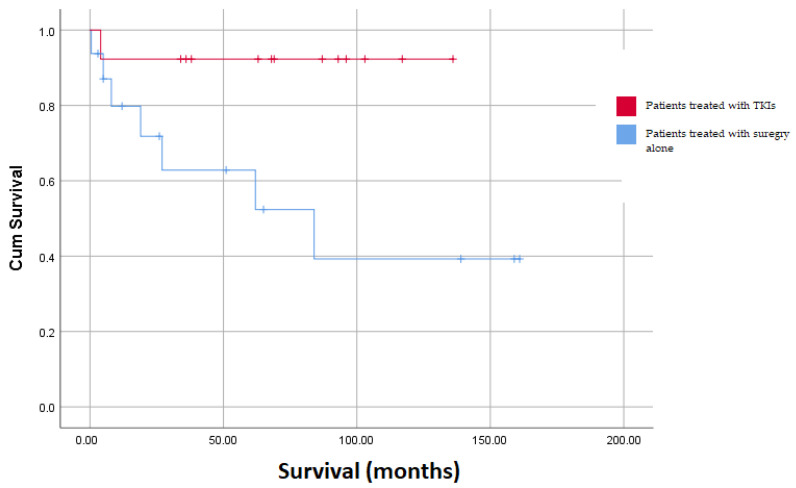
Survival analysis between patients treated with adjuvant therapy and patients treated with surgery alone.

**Table 1 jpm-13-01254-t001:** Macroscopic, histological, and immunohistochemical characteristics of the tumors.

N. ^1^	Age ^2^	Gender	Location	Size ^3^	SMA ^4^	CD117	CD34	Mitotic Rate (HPFs ^5^)	Adjuvant Therapy	Survival Time ^6^	Recurrence
1	65	female	Stomach	4.2	+	+	+	0–1	No	3.0	No
2	77	male	Stomach	15.3	+	+	+	69	No	5.0	No
3	75	female	Stomach	0.5	−/+	+	+	11	No	19.0	No
4	27	male	Stomach	4.9	−/+	+	+	8	No	26.0	No
5	54	female	Stomach	11.5	−/+	+	+	4	Yes	38.0	No
6	49	male	Stomach	2.5	−	+	+	0	No	12.0	No
7	45	male	Stomach	0.9	−	+	+	2	Yes	34.0	No
8	54	female	Stomach	3.8	−	+	+	16	No	161.0	No
9	56	male	Stomach	4.0	−	+	+	5	No	159.0	No
10	61	male	Stomach	3.8	−	+	+	12	No	139.0	No
11	70	female	Stomach	5.8	−	+	+	30	Yes	4.0	No
12	64	female	Stomach	10.0	−	+	+	15	No	0.50	Yes
13	73	male	Stomach	4.0	−	+	+	2	No	62.0	No
14	64	male	Stomach	3.5	−	+	+	5	Yes	69.0	No
15	65	female	Stomach	6.4	−	+	+	2	No	5.0	Yes
16	73	male	Stomach	4.8	−	+	+	5	No	27.0	No
17	28	male	Stomach	8.6	−	+	+	21	Yes	63.0	No
18	41	male	Stomach	4.7	−	+	+	2	No	65.0	No
19	66	female	Stomach	10.5	−	+	+/−	8	Yes	87.0	No
20	75	female	Stomach	11.0	−	+	−	6	Yes	68.0	No
21	82	female	Stomach	4.8	−	−	+	10	No	8.0	No
22	76	male	S. intestine ^7^	3.2	+	+	+	2	Yes	N/A ^8^	N/A ^8^
23	49	male	S. intestine ^7^	7.4	+	+	+	2	Yes	93.0	No
24	61	female	S. intestine ^7^	11.5	+	+	+	60	Yes	96.0	No
25	60	female	S. intestine ^7^	9.2	−/+	+	+	7	No	84.0	No
26	55	male	S. intestine ^7^	4.4	−/+	+	+	5	Yes	117.0	No
27	71	female	S. intestine ^7^	3.2	−/+	+	+	4	Yes	136.0	No
28	61	female	S. intestine ^7^	12.1	−	+	−	2	Yes	103.0	No
29	63	male	S. intestine ^7^	4.0	−	+	−	5	No	N/A ^8^	N/A ^8^
30	67	male	Left ischiorectal fossa	3.6	−	+	+	5	No	51.0	No
31	38	female	Omentum	6.2	−	+	+	69	No	N/A ^8^	N/A ^8^
32	43	female	Pelvis	4.2	−	+	+	11	No	36.0	No

^1^ N: number of the case, ^2^ age in years, ^3^ size in centimeters, ^4^ SMA: smooth-muscle actin, ^5^ HPFs: high-power fields, ^6^ survival time measured in months and calculated from the day of the operation until the day of our contact with the patients, ^7^ S. Intestine: small intestine, ^8^ N/A: the patient did not answer our call.

**Table 2 jpm-13-01254-t002:** Risk of metastasis (%) based on the mitotic rate, the size, and the location of the tumor (adopted from Miettinen and Lasota 2006).

Mitotic Rate (HPF ^a^)	Tumor Size (cm)	Location of the Tumor			
		Stomach	Duodenum	Jejunum/Ileum	Rectum
≤5/50	≤2	None (0%)	None (0%)	None (0%)	None (0%)
	>2 ≤5	Very low (1.9%)	Low (4.3%)	Low (8.3%)	Low (8.5%)
	>5 ≤10	Low (3.6%)	Moderate (24%)	High (34%) *	High (57%) *
	>10	Moderate (10%)	High (52%)	High (34%) *	High (57%) *
>5/50	≤2	None *	High *	Insufficient data	High (54%)
	>2 ≤5	Moderate (16%)	High (73%)	High (50%)	High (52%)
	>5 ≤10	High (55%)	High (85%)	High (86%) *	High (71%) *
	>10	High (86%)	High (90%)	High (86%) *	High (71%) *

* Very small number of cases. ^a^ High-power field.

## Data Availability

The data that support the findings of this study are available on request from the corresponding author, G.T.

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
