# Peer review of "Gastrointestinal Stromal Tumors: Our Ten-Year Experience of a Single-Center Tertiary Hospital"

_jpm, 2023, doi:10.3390/jpm13081254_

Round 1

Reviewer 1 Report (Previous Reviewer 3)

The Authors did not adequately answer my requests 2 and 3

Author Response

First of all, we would like to thank the reviewer for these valuable comments. We have clarified and analyzed the parts that were not adequate. The new answers are highlighted in yellow.  

Reviewer: "In my opinion, it could be interesting to describe also the outcome of those patients with histological diagnosis of GIST (performed endoscopically) who did not undergo resection."

1st Reply: The data requested do not exist, because as a surgical department, they were referred to as only those patients who needed surgical treatment.

To clarify even more, the patients that did not undergo resection, they never reached our department because after the endoscopic diagnosis of GIST they referred to oncology deparment for further conservative treatment. So it is not possible for us to know their outcome.

Reviewer: "Similarly, endoscopic findings should be described. Particularly, how the preoperative diagnosis was made.  Doing so, discuss the results of two recent papers (PMID: 35915956 and PMID: 35264310)"

1st Reply: Endoscopic findings refer only to cases of the stomach where the diagnosis or suspicion of it can be made endoscopically. This is impossible for small bowel or pelvic tumors. Furthermore, gist as extra-mucosal tumors and so are not always visible under endoscopy. Therefore, the diagnosis was based mainly on imaging criteria.

These two recent papers, prove that  EUS-FNB performed by wet aspiration provides a higher tissue biopsy sampling rate compared to the slow traction technique; however, no statistically significant difference was observed between the two techniques.

Furthermore, EUS-FNB is superior to bite biopsy, both in terms of diagnostic performance and safety.

Reviewer 2 Report (New Reviewer)

In table 1, the authors should include (1) who had adjuvant therapy, (2) who had recurrence, and (3) survival time. 

I would like to see the Kaplan-Meier comparing overall or progression free survival of patients with vs. without imatinib.  

What are new findings of this study, anyway?  

Minor English editing is required. 

Author Response

We thank the reviewer for her/his comments.

We have added the information regarding (1) who had adjuvant therapy, (2) who had recurrence, and (3) survival time in Table 1. 

Moreover, we made the survival analysis between patients recieving TKIs and patients treated with surgery alone. The Kaplan-Meier curve is presented in Figure 4.

Finally, our study aimed to describe the experience of our tertiary surgical department on the diagnosis, evaluation, and treatment of GISTs and to provide more information in the beforementioned field about the situation in Greece which lacks significantly updated data.

Round 2

Reviewer 1 Report (Previous Reviewer 3)

The Authors didn't answer to my requests.

Reviewer 2 Report (New Reviewer)

The revised manuscript has been improved. 

No problem

This manuscript is a resubmission of an earlier submission. The following is a list of the peer review reports and author responses from that submission.

Round 1

Reviewer 1 Report

Gastrointestinal stromal tumors (GIST) are the most common stromal malignancy and a model of successful and rational development of targeted tumor therapy. The introduction of tyrosine kinase inhibitors with anti-KIT /PDGFRA activity at local and advanced stages significantly improved survival in diseases previously thought to be resistant to all systemic therapies. These guidelines, developed jointly by the Spanish Society of Medical Oncology (SEOM) and the Spanish Sarcoma Research Group (GEIS), provide a multidisciplinary and up-to-date consensus on the diagnosis and treatment of GIST patients.

The authors analyzed the diagnosis, evaluation, and treatment of GISTs, and also analyzed clinical presentation, prognosis, and molecular characteristics. The topic is interesting. However, my major concern is the innovation of the research. My detailed concerns are shown below.

Question 1:

At present, the incidence of gastrointestinal stromal tumors is increasing, treatment is more standardized than before. Growing evidence has confirmed that the use of 26 of TKIs has led to an improvement in patient survival rate. Although this study reviewed 10 years of data from a single center, the number of cases was only 32, and the conclusion was similar to previous literature. There are no new ideas or results.

 Question 2:

Many references are out of date and do not reflect current developments. It is recommended to replace according to the content of the article, preferably with nearly 5 years old.

The manuscript was well written.

Reviewer 2 Report

Thank to the editors for the opportunity to review this article. It describes the single center experience of the treatment of gist. It is interesting, but I dont think it provide something new. Additionally, some points are missing:

1. What way the diagnosis were made? Were the patients symptomatic?

2. There is lack of important data in charactrristics - what was the BMI of patients? Any comorbidities? Previous surgeries? It would be interesting if you would estimate some risk factor of gist. 

3. What surgeries were performed - lap or open? Why?

4. The first part of the discussion is more results than the real discussion. The authors do not compare their results with the literature. Moreover if the discussion is a review of literature- it is not enough. 

The paper needs Major revisions.

Reviewer 3 Report

This is a retrospective study conducted in a single center, including all patients with GIST who underwent surgical resection between 2008 and 2018. 

Title: should be changed because you can't talk about the "Greek condition" based on a single-center experience. Differently, the presence of a literature review should be mentioned in the title.

- In my opinion, it could be interesting to describe also the outcome of those patients with histological diagnosis of GIST (performed endoscopically) who did not undergo resection.

- Similarly, endoscopic findings should be described. Particularly, how the preoperative diagnosis was made. Doing so, discuss the results of two recent papers (PMID: 35915956 and PMID: 35264310) 

- The results of the literature review should be reported in the "results" section and discussed in the "discussion" section